# Ultra-Wideband and High-Gain Vivaldi Antenna with Artificial Electromagnetic Materials

**DOI:** 10.3390/mi14071329

**Published:** 2023-06-29

**Authors:** Ruiyue Hu, Feng Zhang, Shengbo Ye, Guangyou Fang

**Affiliations:** 1Aerospace Information Research Institute, Chinese Academy of Sciences, Beijing 100190, China; huruiyue21@mails.ucas.ac.cn (R.H.); sbye@mail.ie.ac.cn (S.Y.); gyfang@mail.ie.ac.cn (G.F.); 2Key Laboratory of Electromagnetic Radiation and Sensing Technology, Chinese Academy of Sciences, Beijing 100190, China; 3School of Electronic, Electrical and Communication Engineering, University of Chinese Academy of Sciences, Beijing 100049, China

**Keywords:** metamaterial, artificial electromagnetic material, ultrawide bandwidth, high gain, ground-penetrating radar, Vivaldi antenna

## Abstract

An ultra-wideband and high-gain Vivaldi antenna with artificial electromagnetic material, suitable for ground-penetrating radar (GPR) systems, is proposed. Directors loaded inside the antenna gradient slot direct electromagnetic waves by inducing current to improve gain. The artificial electromagnetic material, also called metamaterial, is composed of multiple “H”-shaped units arranged in a certain regular pattern, loaded at the antenna aperture. The artificial electromagnetic units affect the antenna radiation waves by changing the refractive index to improve radiation directivity. The four Vivaldi units are arranged into a horn-shaped array, and each two units are orthogonally fed to realize dual polarization. Experimental results demonstrate that the antenna has good impedance matching of $S_{11}\leq -10$ dB in 0.9–4 GHz, and the maximum realized gain can reach 15.2 dBi.

## 1. Introduction

The demand for GPR technology is becoming increasingly urgent due to the growing requirement for underground resource development and underground target detection. GPR is widely used in the detection of hidden targets such as underground cavities, bridge defects, underground pipelines, and archaeology, due to its advantages of being non-destructive, fast and high resolution. GPR uses the penetrating property of electromagnetic waves to detect targets underground and behind walls. The higher the frequency of the electromagnetic wave, the higher the loss when penetrating the ground and walls. Thus, a GPR system mostly uses a lower operating frequency [1]. In order to meet the requirement of resolution, the working bandwidth of GPR is generally wide. Due to the requirements of the working frequency and bandwidth, the design of the GPR antenna has become particularly challenging. The ultra-wideband antenna is an important part of GPR system, and its performance determines the radar’s ability to transmit signals and receive echoes. Currently, a variety of UWB antennas have been applied in GPR, including the Vivaldi antenna, horn antenna, and non-frequency-variable antenna. Among them, the Vivaldi antenna is favored due to its advantages of wide bandwidth, compact structure and ease of manufacture.

Ground-penetrating radar antennas not only need to meet the requirements of UWB, but also require strong directivity. The directivity of an antenna refers to its ability to radiate and receive differently in different directions. Ground-penetrating radar needs to detect underground targets, and in order to minimize interference, high-directional antennas should be used [2]. Antenna gain plays a critical role in determining the effective range of the antenna, and as the gain of the antenna increases, the range of its effectiveness increases as well. In order to improve the detection depth of GPR, it is necessary to maximize the antenna gain as much as possible. Increasing the antenna gain can also reduce the power requirement of the transmitter in the GPR system.

Up to now, there have been many methods that improve the bandwidth and gain of the UWB antenna for a GPR system. For example, substrate-integrated waveguide (SIW) is loaded onto a dielectric substrate [3]. SIW is similar to a slotted metal waveguide, but it has the advantage of low return loss compared to other transmission lines. The integrated waveguide structure can improve the gain of the antenna by sacrificing a certain bandwidth. The gain of the antenna can also be enhanced by adding a directional director [4], which concentrates the electromagnetic waves of the antenna aperture to the position of the director. Using an elliptical coupler and irregularly spaced slots can reduce side lobe radiation [5]. A tapered slot structure with a band-resonant cavity is also used to improve bandwidth and gain [6]. By slotting on the dielectric substrate [7], the electromagnetic waves propagate on the substrate between the antenna slot gaps, thereby improving the gain. The miniaturized Vivaldi antenna [8] uses bent feed lines and Gaussian curves to save space, and it replaces the traditional exponential function with a Gaussian function to increase the current path and to improve the port impedance characteristics. The current path can be extended by slotting on the edge of the main radiant surface [9], achieving miniaturization and reducing the low-frequency operating frequency. Double-slot podal and antipodal UWB antennas are designed with out-of-phase uni-planner power dividers [10] to improve bandwidth and gain. A 1 × 4 antenna array [11] consisting of edge-slotted antipodal Vivaldi antenna units arranged equidistantly on the vertical plane achieves an ultra-wideband of 25:1 and a wide-angle scanning characteristic of 30 degrees. A compact antenna [12] is designed through a shared aperture structure, which lowers the minimum operating frequency and improves port isolation.

In recent years, artificial electromagnetic material, known as metamaterial, has been widely used to enhance antenna directivity, improve antenna gain and concentrate the antenna beam. Artificial electromagnetic material is formed by the periodic or aperiodic arrangement of artificial units whose sizes are sub-wavelength, which can be equivalent to real electromagnetic material in a macro sense. Theoretically, it can achieve an arbitrary dielectric constant and permeability, such as zero refractive index and negative refractive index [13]. Metamaterials, such as artificial dielectric lens and FSS reflectors, can meet the bandwidth requirements of ground-penetrating radar ultra-wideband antennas. The gain of ultra-wideband antennas can also be enhanced by strengthening forward radiation, such as artificial dielectric lens, or by reflecting the backward radiation, such as FSS reflectors [14,15], to achieve high-gain requirements. With the application of artificial electromagnetic material in antenna design, there are various ways to further improve antenna performance. For example, metasurface lens are formed by artificial electromagnetic materials [16], which improve radiation directionality without affecting antenna radiation efficiency and return loss. In order to improve directivity, a zero-refractive index material composed of curved lines is loaded into the front section of the Vivaldi antenna [17]. Artificial electromagnetic materials can be used not only to improve directivity, but also to improve isolation between the two antenna elements. For example, the Mu-negative metamaterial is used as a filter [18], allowing the antenna to be compact while maintaining good isolation. In multi-band antennas [19], the isolation between isolated antenna elements is enhanced by loading metamaterials between closely spaced antenna elements in the lowest and highest frequencies.

In this paper, based on the shared aperture dual-polarized Vivaldi antenna, in order to further improve the gain, the following two methods are adopted:A four-element director is loaded into the gradient slot of the Vivaldi antenna unit, and the electromagnetic wave is directed to the edge of the director to enhance the middle and high frequency gain;“H”-shaped metamaterial units are designed and loaded into the aperture of the Vivaldi antenna in a periodic arrangement to make the beam more concentrated by changing the refractive index.

By combining all of the above methods, the gain can be increased to 15.2 dBi without affecting the bandwidth, and the working frequency is between 0.9 and 4 GHz.

## 2. Antenna Design and Methods

### 2.1. Antenna Configuration

The configuration of the high-gain dual-polarization Vivaldi antenna is shown in Figure 1. Inspired by reference [12], the antenna is arranged in a shared aperture array to maintain port isolation and to improve bandwidth. A quadruple director is loaded into the center of the Vivaldi antenna’s tapered slot to increase its gain. At the aperture of the antenna, a 4×15 array of artificial electromagnetic materials is loaded. The metal patches, arranged periodically, can be equivalent to a medium with a refractive index greater than 1, thus realizing the function of a lens. The model of a single Vivaldi element and its various dimensions are shown in Figure 1a on the front and in Figure 1b on the back. The exponential curve expression used for the tapered slot of the Vivaldi unit is given by:(1)$$x=12.28e^{0.018y}-2.48.$$

A quadruple director with a size of approximately $0.33\lambda_{min}$ is loaded into the middle part of the tapered slot. The substrate used for the antenna is Rogers RT5880, with a relative permittivity of 2.2 and a thickness of 1.5 mm. The final dimensions of the antenna are 240 × 240 × 289 mm${}^{3}$. Table 1 lists the values of all the parameters.

The antenna designed in this paper can maintain a good return loss at 0.9–4 GHz, lower than −10 dB, and achieve a gain of 15.2 dBi by loading the directors and metamaterials.

### 2.2. Working Principle

#### 2.2.1. Director

The director is loaded inside the tapered slot of the initial Vivaldi antenna, which increases the gain of the antenna without changing its size. When the Vivaldi antenna is fed, the radiation electric field produced by the antenna points from one side of the tapered slot to the other side, that is, the radiation electric field is along the x-axis. The electric field of the antenna obtained by CST Studio Suite (CST) software simulation is shown in Figure 2. The director structure can be viewed as a metal wire. When the director is inserted onto the substrate in the middle of the tapered slot, the induced current is generated on the director due to the parallel orientation of the electric field and the director, which plays a role in directing the radiation electric field, as shown in Figure 3.

When the Vivaldi antenna operates, the feeding current is located between the tapered slot lines and propagates toward the aperture along the slot lines. Due to the short distance between the slot lines near the feeding end, which is less than half a wavelength of the antenna’s operating frequency, the electric field transmits toward the antenna aperture within the substrate without radiating outward. As the current travels, when it reaches the slot line with a spacing of half the operating wavelength of the antenna, electromagnetic waves begin to radiate. At this point, the antenna can be equivalent to a radiation source, generating radiation electromagnetic waves. When the length of the director is shorter than half the operating wavelength, the induced electromagnetic waves generated by the induced current of the director are phase-delayed compared to the radiation electromagnetic waves, thereby directing the radiation. Conversely, when the length of the director is longer than half the operating wavelength, the induced electromagnetic waves generated by the director are phase-advanced and will reflect the radiation. The position of the director also affects the antenna’s performance. When the length of the director is less than half the operating wavelength and is located close to the radiation source, it plays a role in directing the radiation. However, when the length of the director is too far away, it will reflect the radiation electromagnetic waves, thereby degrading the antenna’s gain.

Based on the above principles, the influence of different lengths of the directors on the antenna gain are compared, as shown in Figure 4, and the length of the director is finally determined to be 25 mm. Additionally, by continuously optimizing the position of the directors, the final structure of the Vivaldi antenna loaded with the directors is shown in Figure 5. Figure 6 shows the comparison between the gain results obtained by simulation with and without the directors. From the simulation results, the gain of the Vivaldi antenna loaded with the directors is significantly improved at high frequency, but the improvement on the low-frequency band is not obvious. Because, at low-frequency, the radiation position of the Vivaldi antenna is close to the aperture or even ahead of the position of the directors, the improvement effect is not obvious.

#### 2.2.2. Artificial Electromagnetic Material

When electromagnetic waves propagate in the air and enter the medium at a certain angle, according to Snell’s law of refraction:(2)$$n_{1}\sin\theta_{1}=n_{2}\sin\theta_{2},$$
where $\theta_{1}$ and $\theta_{2}$ are the incident and refracted angles of the electromagnetic wave, and $n_{1}$ and $n_{2}$ are the refractive indices of the two media. When the refractive index of the medium, $n_{2}$, is greater than that of air, $n_{1}$($n_{1}=1$), according to Formula (Equation 2), it can be seen that the refracted angle, $\theta_{2}$, is greater than the incident angle, $\theta_{1}$. Therefore, the electromagnetic waves can be focused toward the normal direction of the interface, thereby achieving the effect of increasing the gain, as shown in Figure 7.

This article presents the design of periodic H-shaped metal patches printed on a 1.5 mm thick Rogers RT5880 substrate. Figure 8 shows the structure of the H-shaped artificial electromagnetic material unit and equivalent circuit diagram. The rectangular loop of the lens unit acts as the inductor $L_{lens}$, and the H-shaped gap acts as the capacitor $C_{lens}$ and finally forms the equivalent series resonant circuit with lumped components. Table 2 provides the values of the various parameters of this unit. The simulation software CST Studio Suite is used for periodic simulation of the artificial electromagnetic material unit, and the S-parameter response of the structure is calculated using a frequency domain solver.

According to the formula proposed in reference [20]:(3)$$n=\frac{1}{kd}\cos^{-1}\left[\frac{1}{2S_{21}}\left(1-S_{11}^{2}+S_{21}^{2}\right)\right],$$
(4)$$z=\sqrt[{}]{\frac{{\left(1+S_{11}\right)}^{2}-S_{21}^{2}}{{\left(1-S_{11}\right)}^{2}-S_{21}^{2}}},$$
where *n* is the effective refractive index, *k* is the free space wave number, *d* is the effective substrate thickness, $S_{11}$ and $S_{21}$ are the simulated S-parameters, and *z* is the wave impedance. Since the S-parameters are complex, the obtained effective refractive index and wave impedance are also complex. The imaginary part of the refractive index represents the absorption of the electromagnetic wave by the medium, while the real part of the wave impedance represents the attenuation of the electromagnetic wave. Since the medium is a passive medium, the imaginary part of the effective refractive index must satisfy $Im\left(n\right)\geq 0$, and the real part of the wave impedance must satisfy $Re\left(z\right)\geq 0$. In order to satisfy the above conditions, Formula (Equation 4) is improved to eliminate the periodic inverse trigonometric function. Assuming that $\Delta =\frac{1}{2S_{21}}\left(1-S_{11}^{2}+S_{21}^{2}\right)$, Formula (Equation 3) can be expressed using Euler’s formula as follows: (5)$$\Delta =\cos\left(nkd-2m\pi \right)=\frac{e^{j\left(nkd-2m\pi \right)}+e^{-j\left(nkd-2m\pi \right)}}{2}.$$

The square root of a complex number has two values: (6)$$\sqrt[{}]{1-\Delta^{2}}=\pm j\frac{e^{j\left(nkd-2m\pi \right)}-e^{-j\left(nkd-2m\pi \right)}}{2}.$$

Formulas (Equation 5) and (Equation 6) can be simultaneously obtained: (7)$$n=-\frac{j}{kd}\ln_{}\left[\Delta \pm j\sqrt[{}]{\left(1-\Delta^{2}\right)}\right]+\frac{2m\pi }{kd},$$
where *m* is the number of periods. By substituting the simulated S-parameters into Formula (Equation 7), the effective refractive index of the artificial electromagnetic materials can be obtained [21,22]. Figure 9 shows the simulated S-parameters, as well as the effective permittivity, effective permeability, effective refractive index, and wave impedance of the artificial electromagnetic material.

It can be seen that the “H”-shaped artificial electromagnetic material undergoes resonance at around 13 GHz and maintains a high stability in the frequency range of 0.5–6 GHz. For the required frequency band of 0.9–4 GHz, the “H”-shaped artificial electromagnetic material is in a non-resonant state, and the imaginary part of the effective refractive index is about 0, while the real part gradually increases from 1.68 to 1.78. Loading the metamaterials in front of the Vivaldi antenna aperture can be used to focus the beam and to improve the gain. The gain improvement effect obtained through simulation is shown in Figure 10.

The simulation results show that the maximum gain of the Vivaldi antenna unit with metamaterials is increased by approximately 1 dB. The results indicate that the artificial electromagnetic material can significantly improve the high-frequency gain, but the effect is not ideal at low-frequency. This is because the artificial electromagnetic material is a sub-wavelength artificial structure, and in order to achieve the same gain improvement effect at lower frequencies, a larger structure is required. Considering the size of the antenna, it was decided to use the dimensions of the lens structure mentioned above.

#### 2.2.3. Wideband Wilkinson Power Divider

As the designed Vivaldi antenna requires four-port feeding, a wideband Wilkinson power divider [23] was designed, which can keep $S_{11}$ below −10 dB and port isolation below −15 dB within the 0.9–4 GHz bandwidth. First, a wideband Wilkinson power divider is designed for one-to-two power division, and then, the three Wilkinson power divider ports are connected together using coaxial lines to achieve the requirement of one-to-four power division.

The substrate used for the designed wideband one-to-two power divider in this article is Rogers RT5880, with a thickness of 0.508 mm, relative permittivity of 2.2, and a size of 54.3 × 40 mm${}^{3}$. Figure 11 shows the structure of the simulation model, with port 1 on the left and ports 2 and 3 on the right. The S-parameter response can be obtained through simulation. The detailed design of the Wilkinson power divider will not be repeated in this article. The S-parameters of the power divider obtained by simulation and measurement are shown in Figure 12b, and the prototype of the proposed antenna is shown in Figure 12a.

## 3. Results and Discussion

The proposed antenna array was fabricated, and the performance was measured using the vector network analyzer (E5063A) and the microwave anechoic chamber. The four Vivaldi units were fixed together using screws and were connected using conductive metal stickers, and the antenna prototype is shown in Figure 13a,b.

The reflection coefficients obtained through simulation and vector network analyzer measurements are shown in Figure 14. The blue solid line represents the simulated reflection coefficient of the Vivaldi array, the green dashed line represents the measured reflection coefficient of a single port without the power divider, and the red dashed line represents the reflection coefficient measured when the input of the power divider is connected to the vector network analyzer. One of the output ends is connected to the antenna port, and the other is connected to the matching resistor. By comparison, it can be seen that the loaded power divider did not have a significant impact on the reflection coefficient. Within the operating frequency range of 0.9–4 GHz, the reflection coefficient after loading the power divider is less than −10 dB. The discrepancy between the simulated S-parameters and the measured values may be due to errors in welding and manufacturing, as well as to the asymmetry of the three power divider welding resistors that caused unbalanced power feeding. In addition, the radiation efficiency of the final antenna array obtained by simulation is shown in Figure 15. It can be seen that loading artificial electromagnetic materials has little influence on the antenna radiation efficiency at low frequency, but it can improve the antenna radiation efficiency at high frequency. The maximum radiation efficiency of the antenna can reach 99% at 1.1 GHz, and the minimum radiation efficiency is 83% at 4 GHz.

The radiation patterns and gain of the antenna were measured using the microwave anechoic chamber and were compared to the simulated results. Figure 16 shows the measured and simulated horizontal and vertical polarization radiation patterns at different frequencies, where the red solid line represents the simulated result, and the blue dotted line represents the measured result. The measured radiation patterns are generally consistent with the simulation results. Figure 17 compares the simulated and measured gains of the antenna, where Figure 17a shows the comparison of the gains for horizontal and vertical polarizations, and Figure 17b shows the comparison of the results for the maximum gain, which is the sum of the gains for horizontal and vertical polarizations. The measured gain is close to the simulated gain, and the maximum gain of the antenna is between 6.7 and 15.2 dBi.

Table 3 compares the bandwidth, size and gain of the antenna proposed in this paper with those of ultra-wideband and high-gain antennas in the literature. By comparison, it can be found that the proposed antenna has higher maximum gain than the existing antennas.

## 4. Conclusions

This paper proposes an ultra-wideband and high-gain Vivaldi antenna for a ground-penetrating radar system. The gain is enhanced by loading directors inside the gradient slot of the Vivaldi antenna and by periodically arranging artificial electromagnetic material in front of the antenna aperture. The four ports of the antenna are connected to Wilkinson power dividers to obtain high port isolation and are fed orthogonally to form dual polarization. The antenna has good impedance matching throughout the entire operating frequency band of 0.9 to 4 GHz, and the realized gain can reach between 6.7 to 15.2 dBi. The resulting antenna is electrically large, with dimensions of 0.72 × 0.72 × 0.86$\lambda^{3}$, where λ represents the wavelength of the lowest operating frequency. The gain can be further improved by loading multi-layer planar artificial electromagnetic material lenses above the array aperture, and this will be researched in the future.

## Figures and Tables

**Figure 1 micromachines-14-01329-f001:**
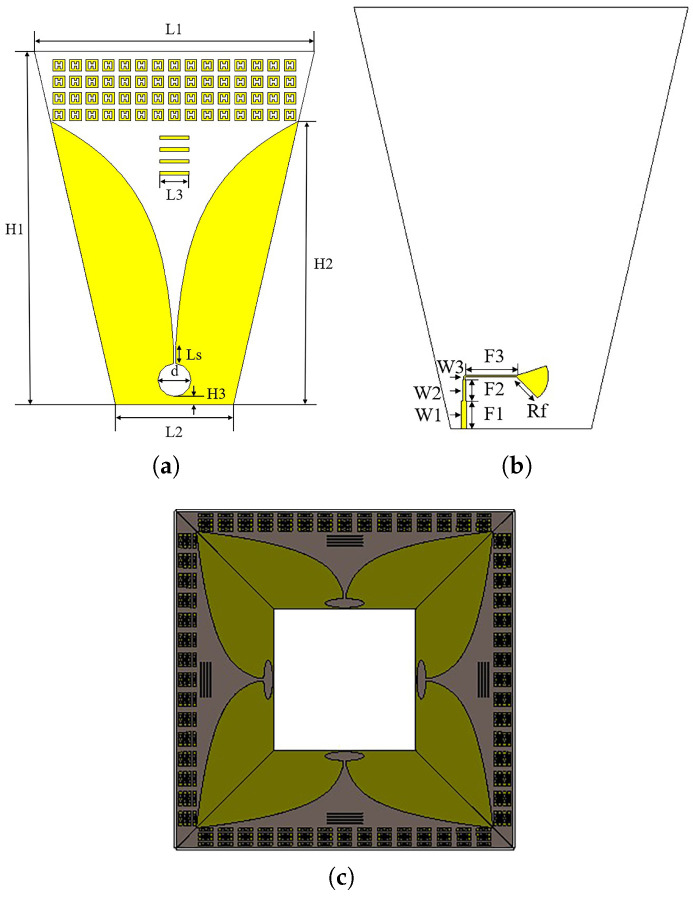
Simulation model of the Vivaldi antenna loaded with metamaterials: (**a**) Front view of a single Vivaldi antenna unit. (**b**) Back view of a single Vivaldi antenna unit. (**c**) Top view of the Vivaldi antenna array.

**Figure 2 micromachines-14-01329-f002:**
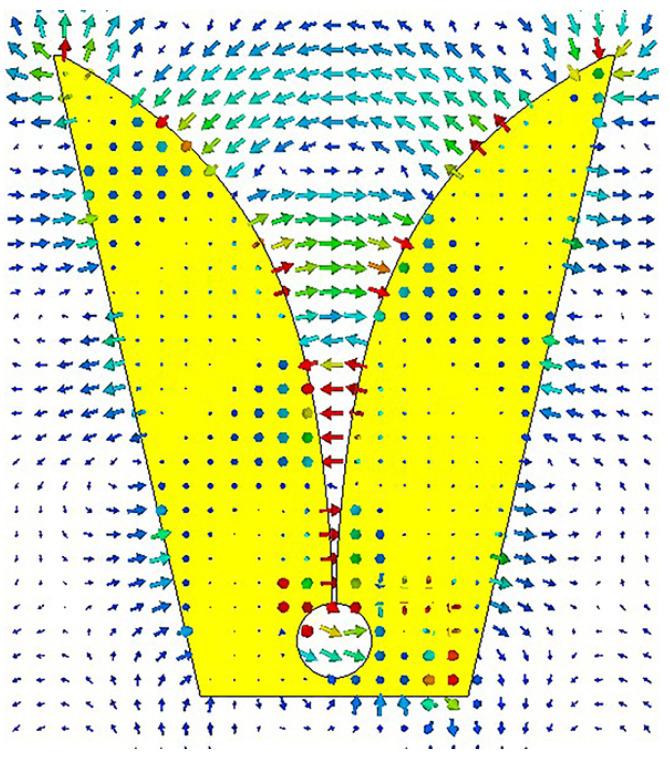
The electric field of the initial Vivaldi antenna.

**Figure 3 micromachines-14-01329-f003:**
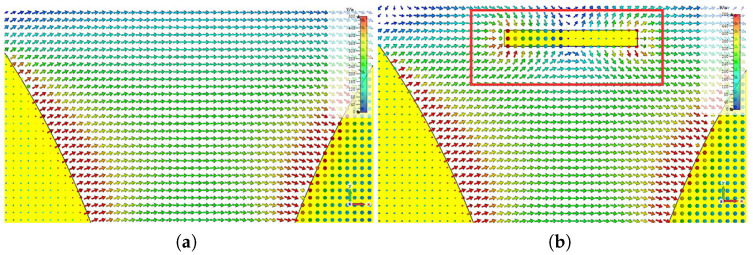
The electric field changes before and after loading the director. (**a**) The electric field simulation before loading the director. (**b**) The electric field changes after loading the director.

**Figure 4 micromachines-14-01329-f004:**
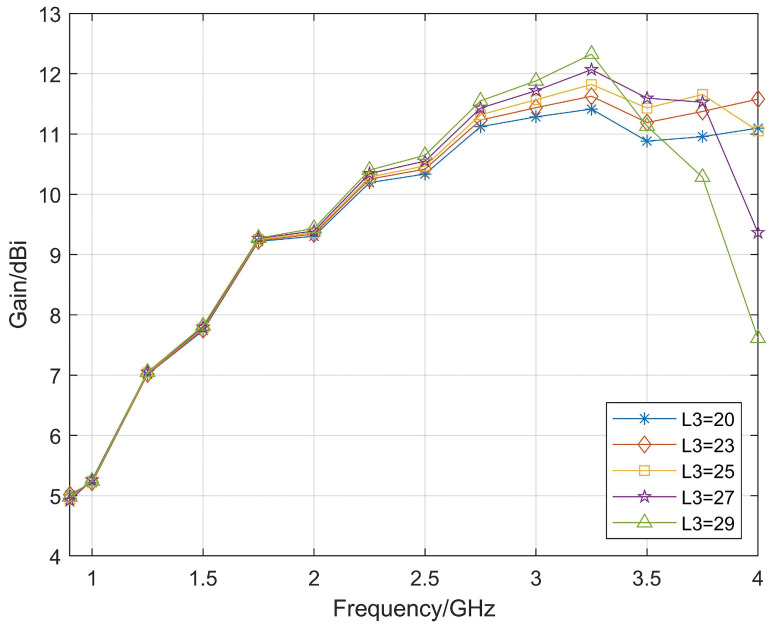
Comparison of antenna with different director lengths.

**Figure 5 micromachines-14-01329-f005:**
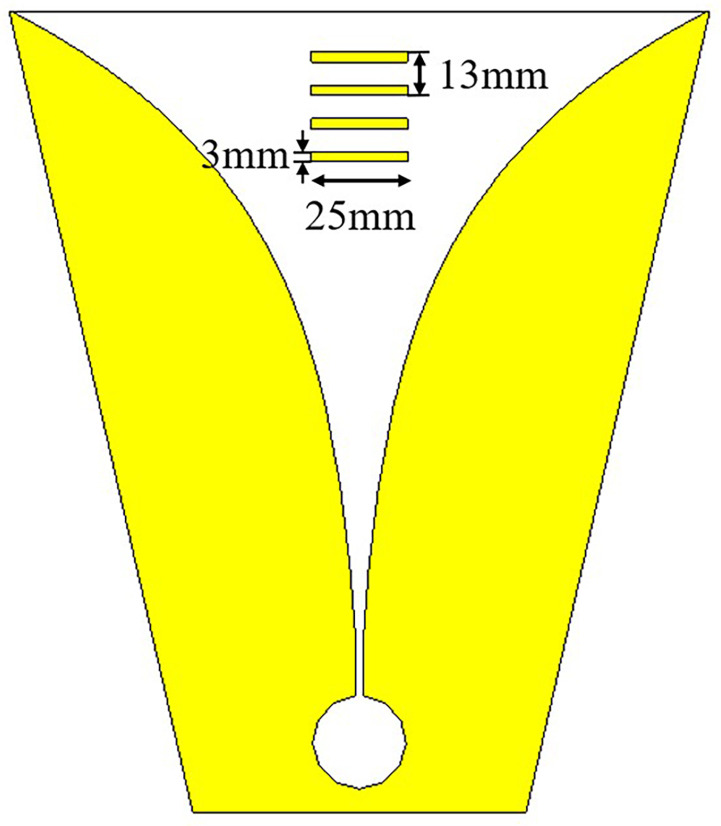
Configuration of Vivaldi antenna element with directors.

**Figure 6 micromachines-14-01329-f006:**
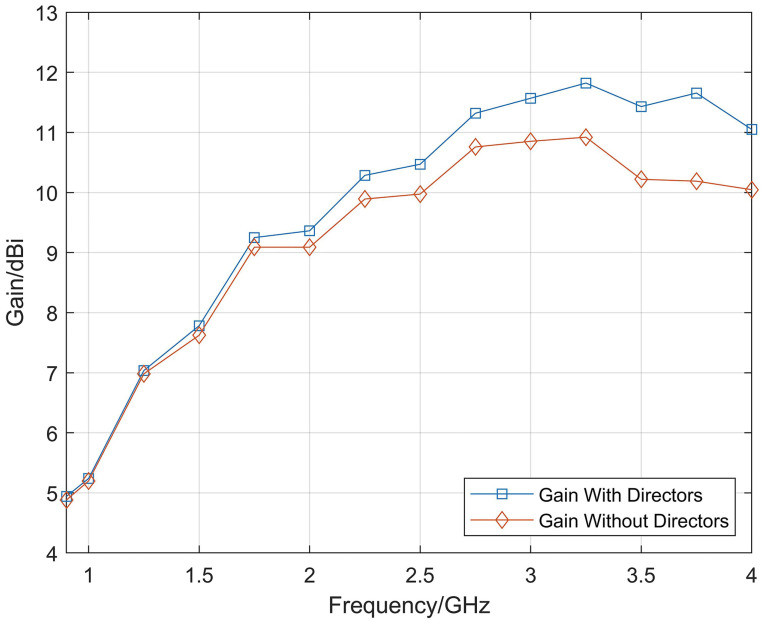
Comparison of Vivaldi antenna gain with initial Vivaldi antenna after loading the directors.

**Figure 7 micromachines-14-01329-f007:**
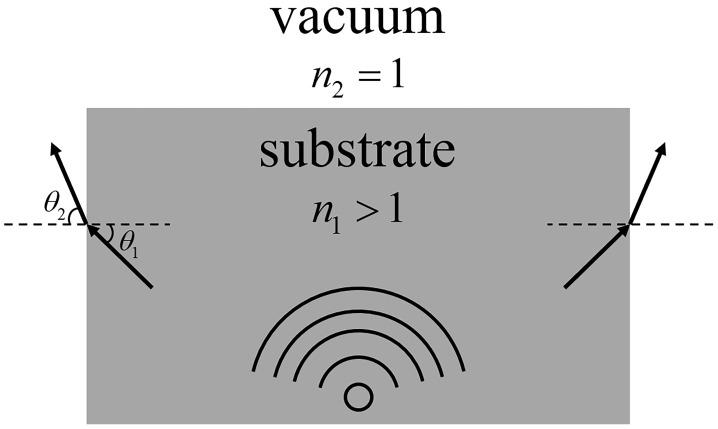
Influence of substrate with refractive index greater than 1.

**Figure 8 micromachines-14-01329-f008:**
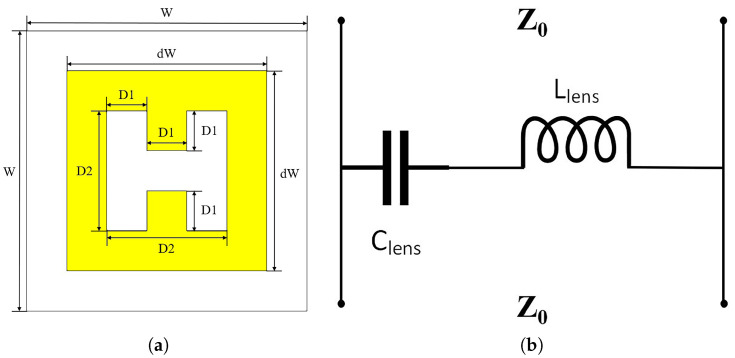
Simulation model. (**a**) Artificial electromagnetic material unit structure diagram. (**b**) Equivalent circuit diagram.

**Figure 9 micromachines-14-01329-f009:**
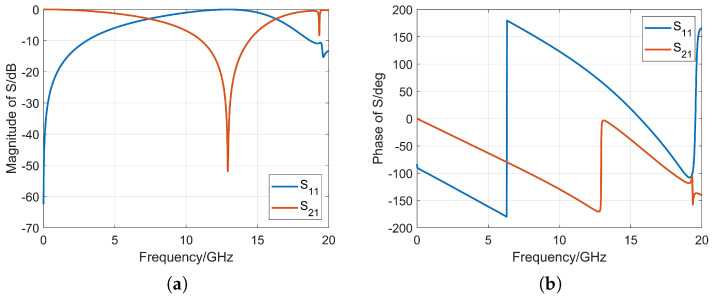
Simulation results of artificial electromagnetic material. (**a**) S-parameter amplitude. (**b**) S-parameter phase. (**c**) Equivalent permittivity. (**d**) Equivalent permeability. (**e**) Equivalent refractive index. (**f**) Equivalent wave impedance.

**Figure 10 micromachines-14-01329-f010:**
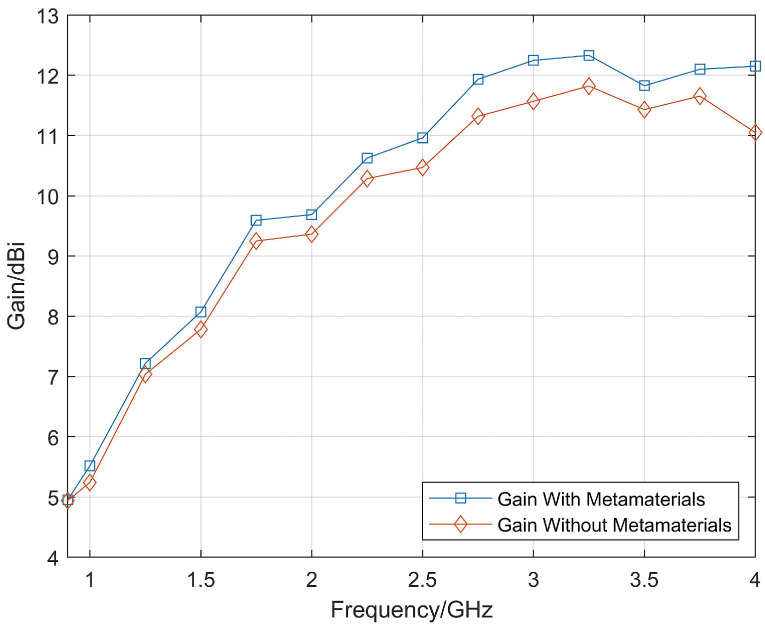
Simulated realized gain with/without metamaterials.

**Figure 11 micromachines-14-01329-f011:**
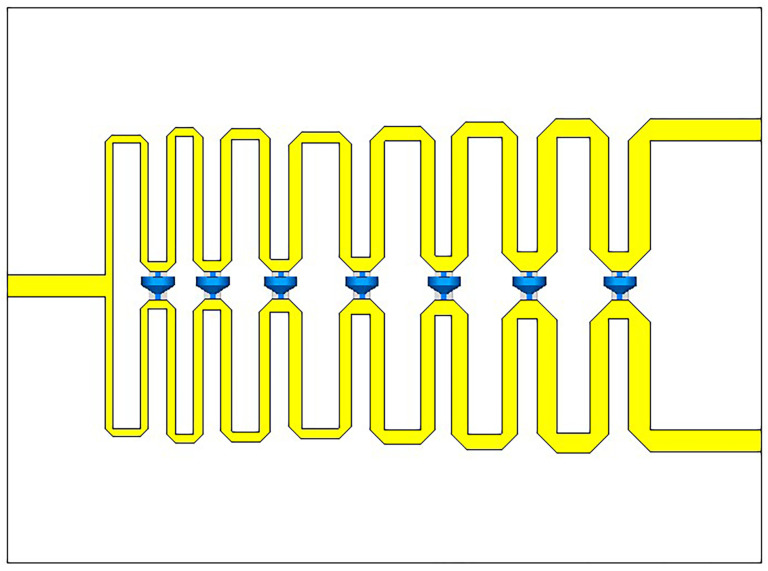
Simulation model of the Wilkinson power divider.

**Figure 12 micromachines-14-01329-f012:**
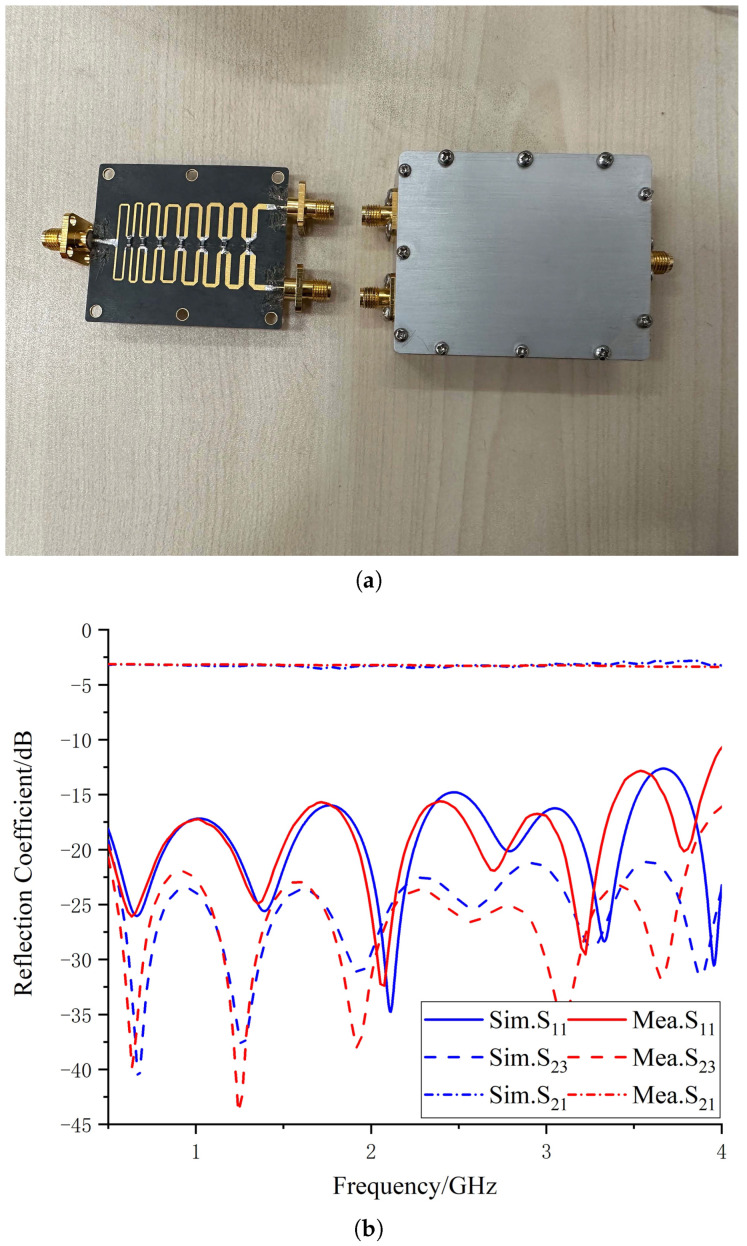
Physical diagram of the Wilkinson power divider and comparison of simulated and measured results. (**a**) Configuration of the Wilkinson power divider. (**b**) Simulated and measured reflection coefficients.

**Figure 13 micromachines-14-01329-f013:**
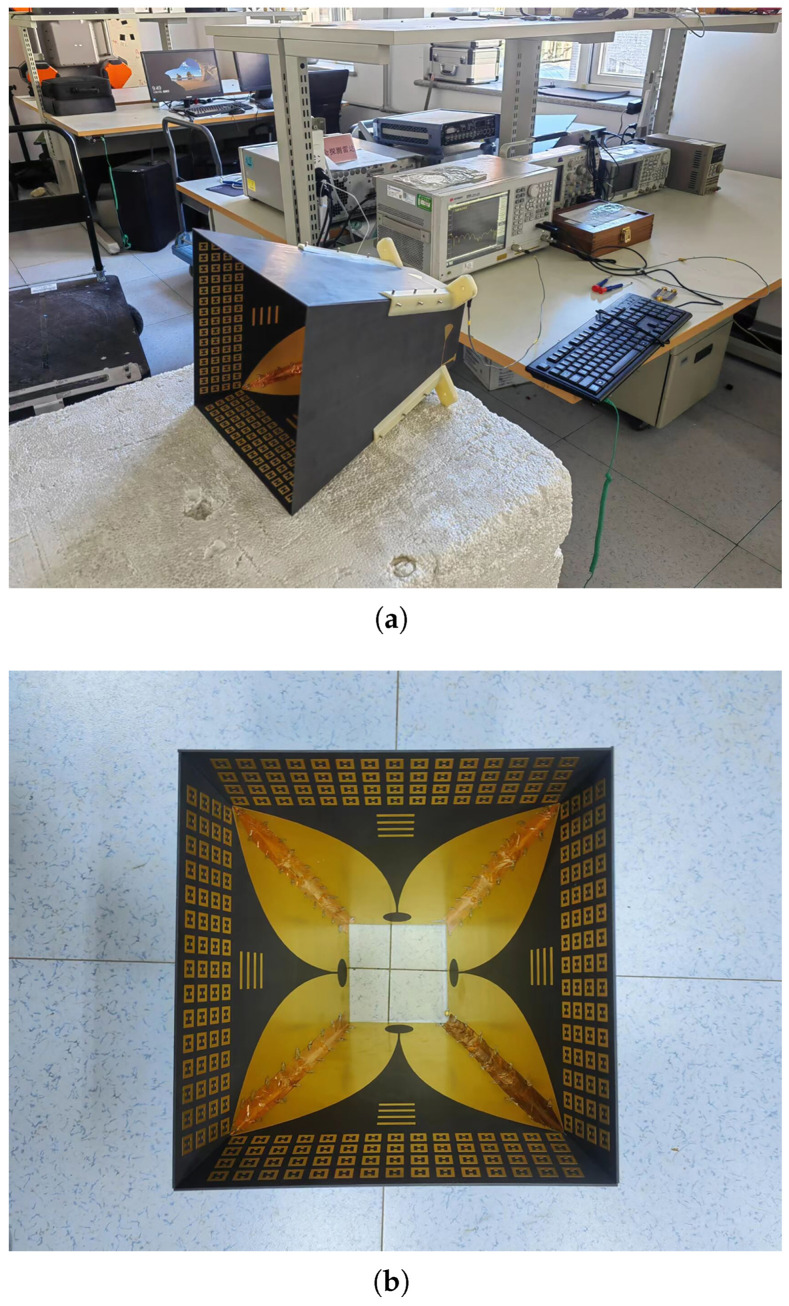
Vivaldi antenna loaded with artificial electromagnetic material. (**a**) Side view. (**b**) Top view.

**Figure 14 micromachines-14-01329-f014:**
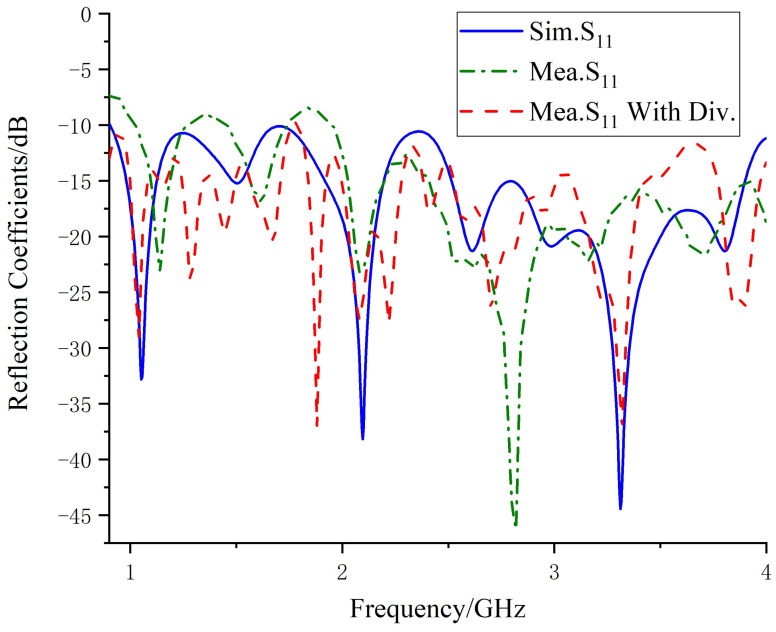
Simulated and measured reflection coefficients.

**Figure 15 micromachines-14-01329-f015:**
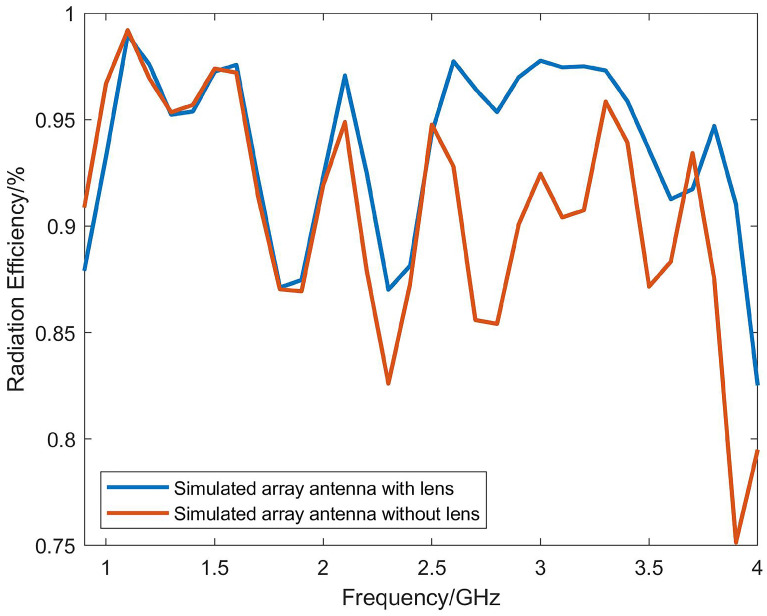
Simulated radiation efficiency.

**Figure 16 micromachines-14-01329-f016:**
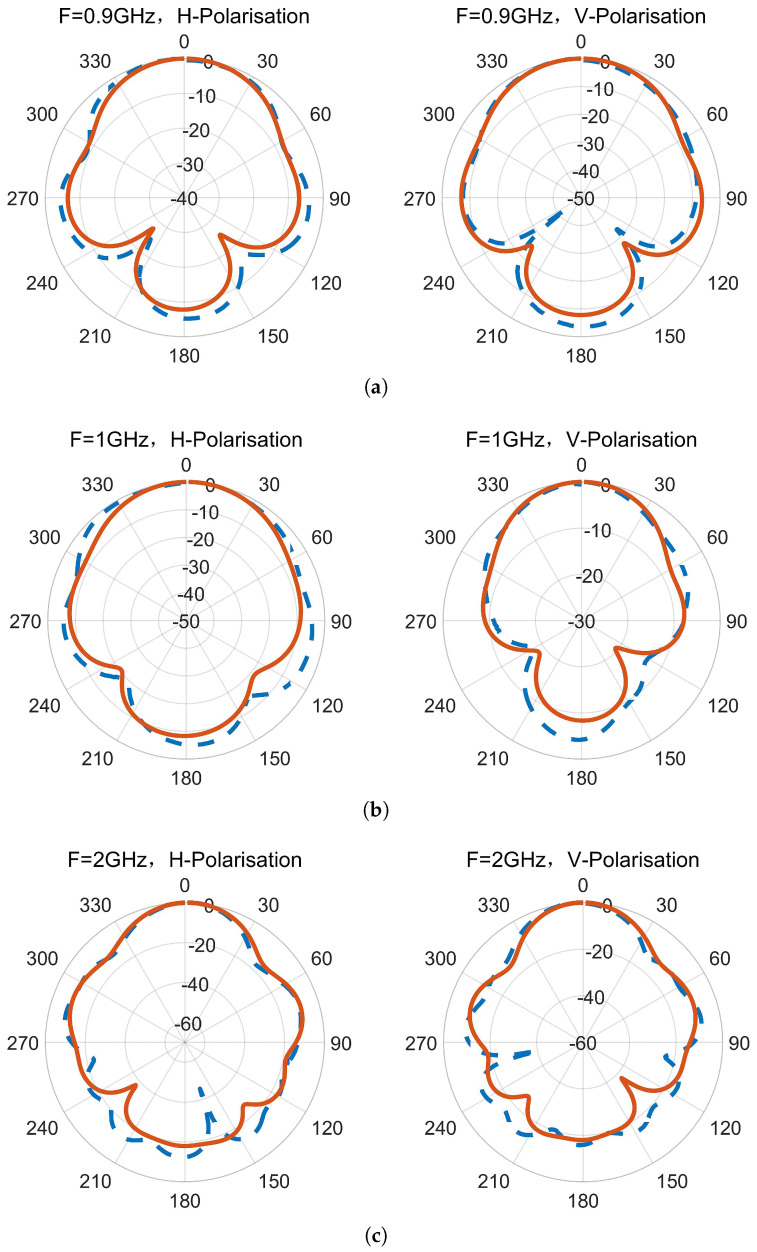
Normalized radiation patterns of H-polarization and V-polarization. (**a**) 0.9 GHz. (**b**) 1 GHz. (**c**) 2 GHz. (**d**) 3 GHz. (**e**) 4 GHz.

**Figure 17 micromachines-14-01329-f017:**
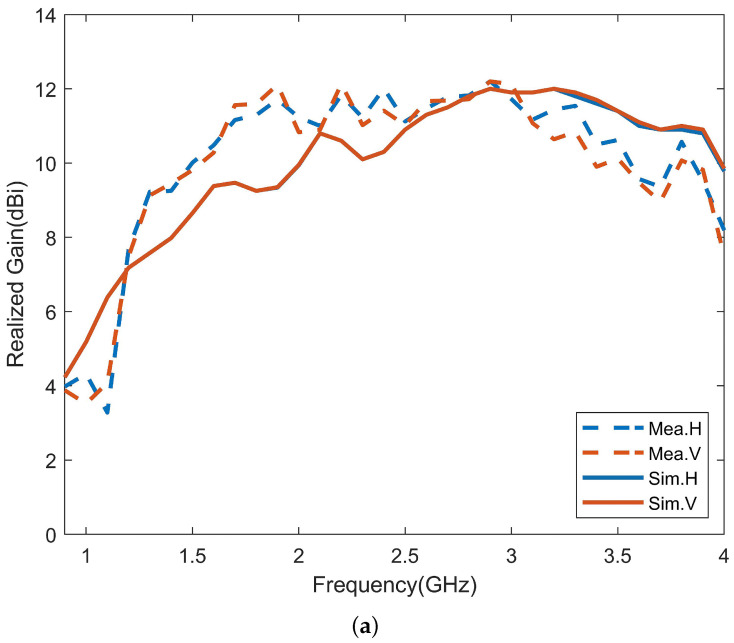
Simulated and measured realized gain of the antenna. (**a**) H-polarization and V-polarization. (**b**) Dual-polarization.

**Table 1 micromachines-14-01329-t001:** Antenna parameters and values (unit: mm).

Paramters	H1	H2	H3	L1	L2	L3	Ls	d	W1	W2	W3	F1	F2	F3	Rf
Values	300	200	7	237.6	100	25	15	28	4	2.3	1.2	20	15	37	21

**Table 2 micromachines-14-01329-t002:** Unit parameters and values (unit: mm).

Paramters	D1	D2	W	dW
Values	2	6	14	10

**Table 3 micromachines-14-01329-t003:** Comparison of antenna performance between the proposed antenna and other works.

Refs.	Freq (GHz)	Size (mm${}^{3}$)	Min. Gain (dBi)	Max. Gain (dBi)
[7]	0.8–6.0	220×256×1.6	2.5	11
[10]	2.4–15.4	70.7×72×0.6	-	11.3
[10]	2.8–16	89.9×74×0.6	-	10.4
[12]	0.4–4.0	210×210×152	4.0	12
[14]	3.5–5.8	160×66×20.1	-	12.4
[15]	3.05–11.9	61×61×10	7.87	9.68
[24]	0.4–2.53	210×216×170	-	11.4
[25]	0.3–2.0	450×600×1.5	4.4	11.5
[26]	0.5–3.0	200×200×180	1.0	8.6
Proposed	0.9–4.0	240×240×289	6.7	15.2

## Data Availability

Not applicable.

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
