# Peer review of "Ultra-Wideband and High-Gain Vivaldi Antenna with Artificial Electromagnetic Materials"

_micromachines, 2023, doi:10.3390/mi14071329_

Round 1
Reviewer 1 Report
The authors demonstrated the Ultra-wideband and High Gain Vivaldi Antenna With Artificial Electromagnetic Materials. The concept is exciting, and the simulation results are reasonably good, showing potentially strong reconfigurability. The antenna gain has been significantly improved when loaded with the Metamaterial reflector. I have the following suggestions before accepting it for publication:
-The Introduction needs improvement. The authors should explain Ultra-wideband antennas in conjunction with metamaterials, such as frequency selective surfaces (FSSs) reflectors, and study their parameters such as gain and bandwidth [1-3]. Additionally, please consider referring to these sources, as they may enhance the value of the Introduction:
[1] Compact Size and High Gain of CPW-Fed UWB Strawberry Artistic Shaped Printed Monopole Antennas Using FSS Single Layer Reflector," in IEEE Access, vol. 8, pp. 92697-92707, 2020, doi: 10.1109/ACCESS.2020.2995069.
[2] Ultra-Wide Band Double-Slot Podal and Antipodal Vivaldi Antennas Feed by Compact Out-Of-Phase Power Divider Slot for Fluid Properties Determination. Sensors 2022, 22, 4543. https://doi.org/10.3390/s22124543.
[3] A Wideband High-Gain Microstrip Array Antenna Integrated with Frequency-Selective Surface for Sub-6 GHz 5G Applications. Micromachines 2022, 13, 1215. https://doi.org/10.3390/mi13081215.
- It would be nice if the authors could provide an equivalent circuit model of the proposed metamaterial (H-shaped).
- I don't understand why the authors provided section 2.2.3 titled 'Wideband Wilkinson Power Divider.' Since the antenna is not designed to detect solid or liquid materials, I suggest removing this section.
- I would like to see the graph depicting the antenna's radiation efficiency.
- Please add more references to Table 3 and compare them with the existing literature review.
That's all for me at this moment! The authors are required to revise the comments above carefully. Thanks
There are minor errors that need to be carefully checked!
Reviewer 2 Report
1. The author’s manuscript said “The artificial electromagnetic material, also called metamaterial, is composed of multiple "H"-shaped units arranged in a certain regular pattern, and is loaded at the antenna aperture.”
Why is the multiple "H"-shaped units of the artificial electromagnetic material?
2. The author’s manuscript said “The four Vivaldi units are arranged into a horn-shaped array, and each two units are orthogonally fed to realize dual polarization.”
Why is the orthogonally and dual polarization of the four Vivaldi units?
3. Please show the software of the Figure 2.
4. Please show the equation of the Figure 4.
5. Please show the reference in Figure 4, Figure 6, and Figure 10.
6. The author’s manuscript said “The results indicate that the artificial electromagnetic material can significantly improve the high-frequency gain, but the effect is not ideal at low-frequency.”
What material is the artificial electromagnetic material?
7. The author’s manuscript said “The four ports of the antenna are connected to Wilkin- son power dividers to obtain high port isolation, and are fed orthogonally to form dual polarization.”
Why is the high port isolation of the Wilkin- son power?
Minor editing of English language required
Round 2
Reviewer 1 Report
Dear authors,
Thank you for taking the time to revise the paper carefully. However, I believe that the article is now ready to be published in its current form.
There are minor errors that need to be carefully checked!